# Primary Pericardial Synovial Sarcoma: A Case Report and Literature Review

**DOI:** 10.3390/diagnostics12010158

**Published:** 2022-01-10

**Authors:** Simona Manole, Roxana Pintican, Emanuel Palade, Maria Magdalena Duma, Alexandra Dadarlat-Pop, Calin Schiau, Ioana Bene, Raluca Rancea, Diana Miclea, Viorel Manole, Adrian Molnar, Carolina Solomon

**Affiliations:** 1Department of Radiology, “Niculae Stancioiu” Heart Institute, 400001 Cluj-Napoca, Romania; simona.manole@gmail.com; 2Department of Radiology, “Iuliu Hatieganu” University of Medicine and Pharmacy, 400012 Cluj-Napoca, Romania; calin.schiau@yahoo.com (C.S.); ioanaboca90@yahoo.com (I.B.); 3Department of Cardiovascular and Thoracic Surgery, “Iuliu Hatieganu” University of Medicine and Pharmacy, 400012 Cluj-Napoca, Romania; paladeemanuel1@gmail.com (E.P.); adimolnar45@yahoo.com (A.M.); 4Department of Thoracic Surgery, Leon Daniello“ Pneumophtysiology Hospital Cluj-Napoca, 400332 Cluj-Napoca, Romania; 5Medimages Breast Center, 400458 Cluj-Napoca, Romania; magdaduma@gmail.com; 6Cardiology Department, Heart Institute “N. Stăncioiu”, 400001 Cluj-Napoca, Romania; dadarlat.alexandra@yahoo.ro (A.D.-P.); raluca_rancea@yahoo.com (R.R.); 7Department of Internal Medicine, Faculty of Medicine, “Iuliu Hatieganu” University of Medicine and Pharmacy, 400012 Cluj-Napoca, Romania; 8Department of Medical Genetics, “Iuliu Hatieganu” University of Medicine and Pharmacy, 400012 Cluj-Napoca, Romania; bolca12diana@yahoo.com; 9Department of Cardiovascular Surgery, Heart Institute “N. Stăncioiu”, 400001 Cluj-Napoca, Romania; v_manole@yahoo.com

**Keywords:** synovial sarcoma, pericardium, intrapericardial mass, superior vena cava syndrome, cardiac tamponade

## Abstract

We report a case of a 52-year-old woman who was referred to our institution with a superior vena cava syndrome and was investigated through echocardiography, CT and MRI revealing a well-defined, encapsulated pericardial mass. The pathology, correlated with the immunohistochemical analysis, concluded it was an extremely rare primary pericardial synovial sarcoma. The patient underwent surgery and chemotherapy with a 16-month disease-free survival and passed away after a contralateral aggressive relapse. Moreover, we discuss the role of each imaging modality together with their pericardial synovial sarcoma reported features.

## 1. Introduction

Primary pericardial tumor is an uncommon, extremely rare disorder. Pleuro-pericardial cyst and lipomas are by far the most common benign pericardial masses, while mesothelioma is the most frequently encountered malignant tumor. 

Primary pericardial sarcomas are a group of extremely rare malignancies which include in order of frequency: angiosarcomas, undifferentiated sarcomas, leiomyosarcomas and, exceptionally, synovial sarcomas. 

To the best of our knowledge, there are less than 15 published cases of primary pericardial synovial sarcomas in the last decade [1,2]. 

Synovial sarcoma (SS) accounts for up to 10% of all soft tissue sarcomas and usually affects older children and young adults. It arises commonly around the knee and the ankle, while the neck, abdomen, pleura and lungs are less common locations [3].

Even if substantial advances were made in terms of SS natural history and behavior, its prognosis is still scarce [4]. SS is considered a high-grade sarcoma with an overall survival rate of 50.8% at 10 years. 

## 2. Case Report

We report a case of a 52-year-old woman who was referred to our institution with a superior vena cava syndrome (dyspnea, jugular vein distension, facial, neck and left upper limb edema). An echocardiography was performed and showed a large intrapericardial mass surrounded by a substantial amount of fluid. Furthermore, cardiac CT and MRI highlighted the mass location and mass effect related with the surrounding mediastinal structures. The large mass was situated anterior to the right side of the heart, close to the superior right pulmonary vein, compressing the superior vena cava, right atrium and right ventricle. A heterogeneous solid mass with late, heterogeneous enhancement was noted on both CT and MRI exams (Figure 1). The tumor seemed to be located within the pericardium, with no coronary artery or cardiac cavity invasion. 

On echocardiography and dynamic sequence MRI (cine sequences), the interventricular septum displayed a paradoxical motion with the right atrium and ventricle being collapsed and dyskinetic. All the above-mentioned characteristics represented indirect signs of cardiac tamponade.

The cardiovascular surgeons opted for an immediate surgical excision without prior biopsy. The surgical approach included a longitudinal pericardiotomy which revealed an encapsulated stiff mass (Figure 2). The tumor appeared to arise at the level of the right pulmonary vein pericardial recess and extended throughout the entire anterior aspect of the right heart. Total tumor resection was possible without intra- or postoperative complications. The excised tissue weighed 300 g and measured 10/8/5 cm.

The pathology report (Figure 3 and Figure 4) described interlacing fascicles of spindle cells with myxoid changes. Tumoral cells displayed a monotonous, uniform pattern with the predominance of rounded, large nuclei. Less than 50% of examined tissue presented necrotic or hemorrhagic areas. Immunohistochemistry revealed a high mitosis index (Ki67 intensely positive) and positive reactivity for the following markers: cytokeratin AE1/3, cytokeratin 7, epithelial membrane antigen (EMA), vimentin, S100, CD99 and Bcl2. Staining for desmin and CD34 was negative. The FISH test for the t (X; 18) (p11.2; q11.2) genetic change proved negative and was used to accurately exclude other differentials. 

The final report concluded that the tumor was a grade 3 monophasic poorly differentiated synovial sarcoma. 

After surgery, the patient was submitted to six cycles of chemotherapy following the MAID protocol (MESNA doxorubicine, ifosfamide, dacarbazine) with acceptable tolerance. 

There were 14 months disease-free with negative follow-up imaging (Figure 5). Four months later, the patient complained of fatigue, chest pain and exertional and at rest dyspnea. She was referred to our department where echocardiography, CT and MRI examinations showed tumoral relapse (Figure 6). The tumor was located adjacent to the left atrium and ventricle, and pulmonary artery trunk, and was extending inferiorly between the right and left pulmonary veins and superiorly towards the inferior surface of the aortic arch (Figure 5). A partial thrombosis of the superior lobar branch of the right pulmonary artery was depicted. The pulmonary veins appeared compressed but permeable. The tumor covered the left circumflex and the anterior interventricular coronary artery. There were no signs of recurrence on the site where the primary tumor was located. Debulking surgery was performed, with incomplete removal of the tumor mass, due to the myocardium and coronary arteries’ involvement. Pathology analysis described the second tumor as a grade 3, poorly differentiated monophasic synovial sarcoma; this time categorized as high-grade with S100 negative and calretinin positive markers, in many more areas than before. The FISH test was again negative for the t(X; 18) (p11.2; q11.2) but positive for *BCL*-2 genetic change. The patient was discharged 10 days after surgery with no notable complications, but unfortunately, she passed over two months later.

## 3. Discussion

Monophasic synovial sarcoma of the pericardium is an extremely rare malignant tumor with challenging diagnosis and an unpredictable outcome. 

One vital differential diagnosis is represented by sarcomatoid mesothelioma, especially in older patients. Distinguishing between the two entities is of utmost importance since the clinical management is completely different [4]. The chromosomal translocation (X; 18) (p11.2; q11.2) is found in almost all (>90%) SS [5] while *BCL*-2 immunohistochemical analysis is found in 0–8% of sarcomatous mesothelioma and a much higher percentage of SS [6,7]. Our patient had two negative FISH tests for the chromosomal translocation t(X; 18) (p11.2; q11.2) but a positive immunohistochemical *BCL*-2 analysis. Pathologists considered that there was enough material examined and the immune-morphologic features were conclusive for a complete diagnosis. 

Recent data report histology and grading as important prognostic factors for synovial sarcoma patients. Furthermore, a worse prognosis was observed for monophasic patients and grade 3 tumors (all *p* < 0.05) [8], as in the case of our patient.

Another important aspect for appropriate surgical management is represented by an accurate report of the tumor boundaries, location and relation with surrounding structures. In the few cases reported in the literature, echocardiography was used as the first-line imaging tool, showing in most of the cases only pericardial effusion. CT and MRI are second-line imaging modalities used to provide more accurate tumor characterization, thus, being able to guide the surgical procedures. 

While the echocardiography usually highlights pericardial effusion and a solid, well-defined mass, the imaging characteristic on CT and MRI are variable. Authors reported masses with ill-defined margins, invading the surrounding structures, with homogeneous or heterogeneous aspect on postcontrast enhancement sequences. On MRI, a mass with slightly hyper-T2 signal intensity was observed and reported by two papers, including our case. 

The role of PET CT in stratifying soft tissue sarcoma patients is still controversial. The latest data report significant differences in SUV uptake among the different subtypes of sarcoma. Synovial and myxoid sarcoma of the extremities seem to have low SUV uptake (<10.3) on PET CT, questioning its role in follow-up, relapsing or recurrent disease [9]. For the pericardial location, the results are contradictory, with max SUV reported between 3–13, ranging from mild to high uptake [10,11]. Thus, the role of PET-CT in the follow-up of patients with pericardial synovial sarcoma, still remains debatable Table 1.

There are a few distinct features in our case when compared to the literature review: first, the lack of invasive aspects—in spite of an aggressive histological subtype of the primary tumor, the mass appeared well defined, encapsulated and occupying only the pericardium, sparing cardiac cavities and blood vessels; second, despite the fact that the primary tumor was very large, the patient survived three times the mean survival length of 6 months compared with the literature reports [13]; third, the pericardial mass was depicted also on echocardiography and last, this is the first reported contralateral aggressive relapse.

Particularly, the margins reported on imaging play an important role in the surgical management of all sarcomas. From a macroscopic point-of-view, MRI is the imaging modality of choice in assessing the tumor irregular surface and invasion in the surrounding structures. Regarding the microscopic evaluation of margins, a “free margin” of <2 cm in width is a practical recommendation for all soft tissue sarcomas [18], but virtually impossible to reach in cases with primary pericardial location.

Our patient presented with well-defined, circumscribed margins on all imaging modality assessed, including MRI; but the peculiar location within the thin pericardium did not allow the pathologist to report any “free-margins”, even if the tumor did not invade the extra-pericardial structures.

## 4. Conclusions

This case report emphasizes the importance of prompt clinical suspicion, accurate histopathological diagnosis and the use of appropriate immunohistochemistry markers in the diagnosis of this rare tumor with an unusual location. Pericardial synovial sarcoma may present with various imaging characteristics on echocardiography, CT and MRI and should be included in the differential diagnosis of well-defined pericardial masses.

## Figures and Tables

**Figure 1 diagnostics-12-00158-f001:**
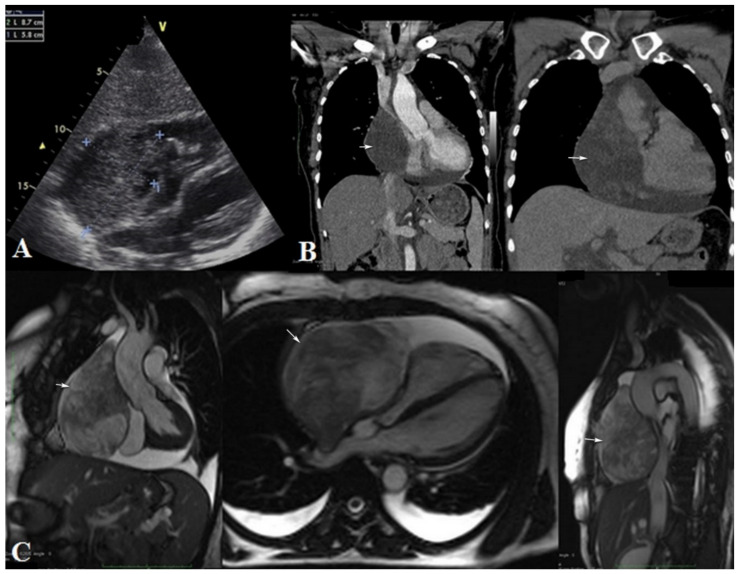
(**A**): Echocardiography; (**B**): computer tomography with contrast enhancement—coronal plane; (**C**): MRI—coronal, axial and sagittal T2 WI; There is a large, inhomogeneous, enhancing intrapericardial mass, surrounded by fluid, with marked compressive effect on the superior vena cava and right heart cavities (arrows). No intracavitary or endoluminal invasive aspects are present.

**Figure 2 diagnostics-12-00158-f002:**
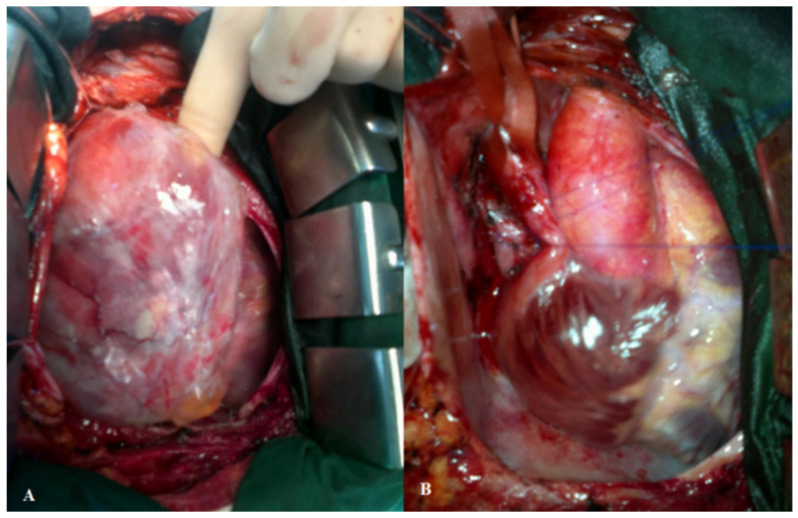
Intraoperative findings: a large, encapsulated mass developed along the entire right aspect of the heart. (**A**): Preoperative image; (**B**): Postoperative image: the SVC can be identified by the blue thread around it.

**Figure 3 diagnostics-12-00158-f003:**
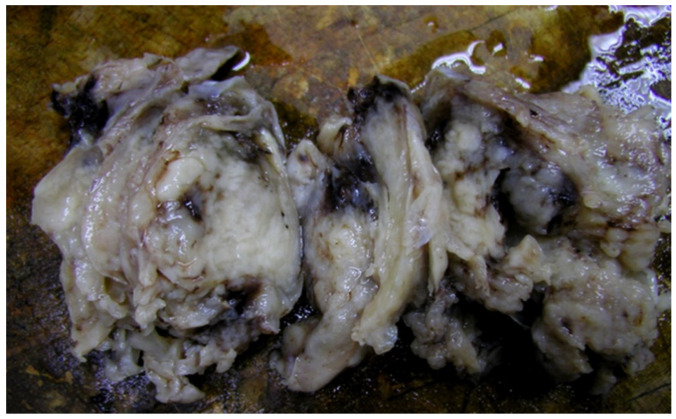
Surgical specimen of the first tumor—gross macroscopic appearance.

**Figure 4 diagnostics-12-00158-f004:**
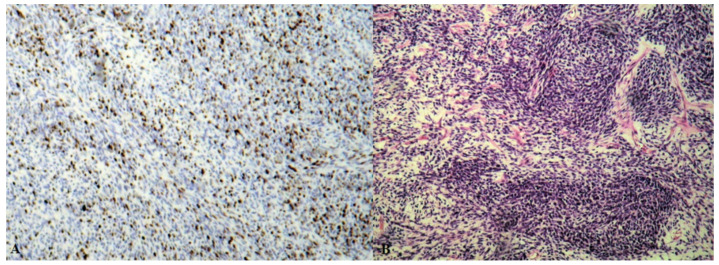
Histopathological findings of the surgical specimen: (**A**): Ki67 immunoreactivity highlights a high-grade sarcoma; (**B**): high-magnification view of the synovial sarcoma; H&E.

**Figure 5 diagnostics-12-00158-f005:**
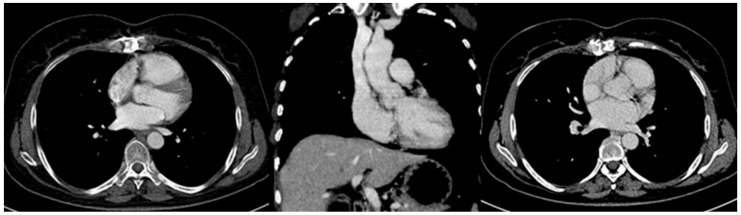
At 14 months, follow-up axial coronal and axial CT images postcontrast enhancement revealed no local relapse.

**Figure 6 diagnostics-12-00158-f006:**
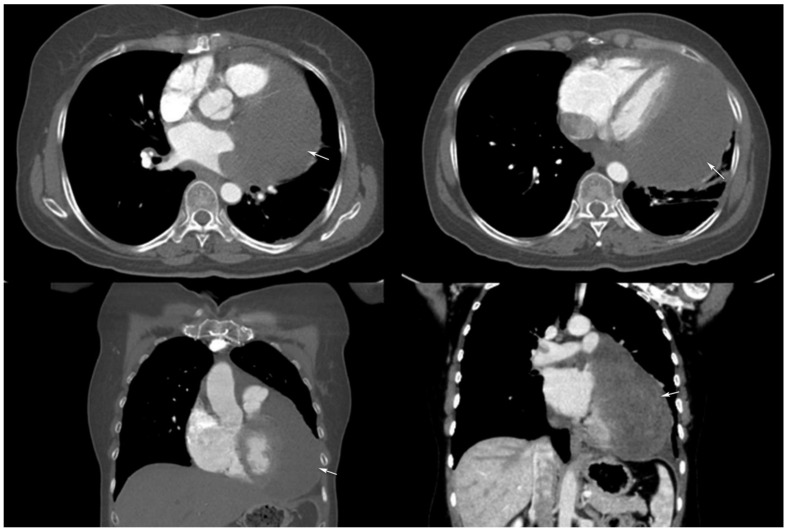
At 18 months, a follow-up CT with contrast enhancement—axial (upper images) and coronal (lower images) plane reveals a homogeneous tumor relapse near the left heart border (arrows), invading the left atrium, right superior pulmonary vein and periaortic root.

**Table 1 diagnostics-12-00158-t001:** Pericardial synovial sarcoma—reported imaging features.

Author/Year	General Aspect	Size (cm)	US *	CT/CT Angiography	MRI	PET CT
Gerry Van der Mieren et al./2004	Recurrent mediastinal mass Ill-defined mass	2.4/2.3	N/A	Mediastinal mass Pleural metastasis	Ill-defined mass Slightly hyper T2 SI	N/A
Gayatri Ravikumar et al./2011	Heterogeneous, invading mass	9/9/2	N/A	Heterogeneous, invading mass	N/A	N/A
Yufan Cheng et al./2012	Heterogeneous, invading mediastinal mass	8/8/2	N/A	Heterogeneous mediastinal mass	N/A	N/A
Prajakta Phatak et al./2014	Heterogeneous pericardial mass	N/A	Pericardial effusion Right atrium collapse	Heterogeneous mass within pericardial cavity	N/A	N/A
Hyo Chul Youn et al./2016	Thickened pericardium	6.3/10.1	Pericardial effusion	Pericardial effusion with thickened pericardium	N/A	N/A
Jose Duran-Moreno et al./2019	Heterogeneous, pericardial, invading mass	8.1/5.6	Pericardial effusion + mass	Mass adherent from the pericardium	Inhomogeneous, mass with cardiac involvement	Mild FDG uptake SUV Max 3.1
Kirsten Y Wong et al./2020	Homogeneous, invading mass	11/10/8	N/A	Homogeneous mass invading coronaries	Myocardium invasion	FDG avid mass SUV max13
Ammar Farook Chapra et al./2021	Heterogeneous, invading pericardial mass	13.5/5	Pericardial effusion	Heterogenous mediastinal mass abutting aorta and heart	Pericardial mass invading pleura and diaphragm	FDG avid mass SUV N/A
Manole Simona et al./2021	Heterogeneous, well-defined, pericardial mass Homogeneous relapse mass	10/8/5	Pericardial mass Pericardial effusion	Heterogeneous, encapsulated mass	Well-defined pericardial mass with iso- and hyper T2 SI	N/A

* US = echocardiography; SI = signal intensity; N/A = not reported; References [6,8,12,13,14,15,16,17].

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
