# Peer review of "Primary Pericardial Synovial Sarcoma: A Case Report and Literature Review"

_diagnostics, 2022, doi:10.3390/diagnostics12010158_

Round 1

Reviewer 1 Report

The topic is interesting and the case report is well described.

Please remove "brief" from the title. An accurate review of the Literature is mandatory.

Regarding FNLCC grade, please cite Bianchi G et al. Eur J Surg Oncol. 2017

Please discuss about PET CT in synovial sarcoma (Sambri A et al Nucl Med Commun. 2019)

A careful and detailed review of the Literature is mandatory.

Conslusions should be further developed. Please suggest how to improve the diagnostic process.

Author Response

The topic is interesting and the case report is well described

Thank you very much

  1. Please remove "brief" from the title. An accurate review of the Literature is mandatory.

Thank you very much for the comment; indeed, our discussion paragraph is not a “brief” review, as it includes all pericardial synovial sarcomas reported until now; we used the term as only few papers are cited; we understand the reviewer perspective, deleted the word “brief” and updated the discussion section.

  1. Regarding FNLCC grade, please cite Bianchi G et al. Eur J Surg Oncol. 2017.

 Thank you for this reference; added the following paragraph and reference: “Recent data reports histology and grading as important prognostic factors for synovial sarcoma patients. Furthermore, a worse prognosis was observed for monophasic patients and grade 3 tumors (all p<0.05)8 as in the case of our patient.

  1. Please discuss about PET CT in synovial sarcoma (Sambri A et al Nucl Med Commun. 2019).

 Thank you for this reference, added; also, we updated the existing paragraph: “PET CT plays an important role in patient’s follow-up, in relapsing or recurrent masses” to: “The role of PET CT in stratifying soft tissue sarcoma patients is still controversial. Latest data reports significant differences in SUV uptake among different subtypes of sarcoma. Synovial and myxoid sarcoma of the extremities seem to have low SUV uptake (< 10.3) on PET CT, questioning its role in follow-up, relapsing or recurrent disease9. For the pericardial location, results are contradictory, with max SUV reported between 3 – 13 with ranging from mild to high uptake10-11. Thus, the role of PET-CT in the follow-up of patients with pericardial synovial sarcoma, still remains debatable.

  1. A careful and detailed review of the Literature is mandatory.

 The discussion section was updated. Thank you for this suggestion. 

  1. Conslusions should be further developed. Please suggest how to improve the diagnostic process.

Thank you for the suggestion; we added the following paragraph: Pericardial synovial sarcoma may present with various imaging characteristics on US, CT and MRI and should be included in the differential diagnosis of well-defined pericardial masses.

Reviewer 2 Report

Dear authors,

I appreciated this manuscript.

I suggest You to expand the Introduction and reference (poorly represented in your paper):

  • Please add in 'Introduction' section a brief paragraph in regards of Soft-tissue sarcomas and the specific subtype 'synovial sarcoma'. (suggested citations PMID: 34687366 - PMID: 34052920)
  • Please discuss briefly, in regards of tumor margins; an important problem for soft-tissue sarcoma rendered more problematic in this delicate position (suggested citation PMID: 33918457)
  • Please expand a little bit more the description of imaging features of tumors. In your schematic literature review add some imaging data (e.g. tumor dimensions).
  • In figure legends add more details about the images showed. Always cite the spatial plane (coronal, sagittal, axial), and exact sequences for MRI (e.g. T1w with contrast media).

Author Response

I appreciated this manuscript.

1. I suggest You to expand the Introduction and reference (poorly represented in your paper): Please add in 'Introduction' section a brief paragraph in regards of Soft-tissue sarcomas and the specific subtype 'synovial sarcoma'. (Suggested citations PMID: 34687366 - PMID: 34052920) .

 Thank you very much for your comment. We included the reference and the paragraph: “Synovial sarcoma (SS) accounts for up to 10% of all soft tissue sarcomas and usually affects older children and young adults. They arise commonly around the knee and the ankle while neck, abdomen, pleura and lungs are less common locations3.

Even if substantial advances were made regarding the natural history and behavior of SS, its prognosis is still scarce4. SS is considered a high-grade sarcoma with an overall survival rate of 50.8% at 10 years”

2.Please discuss briefly, in regards of tumor margins; an important problem for soft-tissue sarcoma rendered more problematic in this delicate position (suggested citation PMID: 33918457)

Thank you for the comment; we added: Particularly, the margins reported on imaging play an important role in the surgical management of all sarcomas13. From a macroscopic point-of-view MRI is the imaging modality of choice in assessing the tumor irregular surface and invasion in the surrounding tissue. Regarding the microscopic evaluation of margins, a “free margin” of < 2 cm in width is a practical recommendation for soft tissue sarcomas.

3. Our patient presented with well-defined, circumscribed margins on all imaging modality assessed, including MRI; but the peculiar location within the thin pericardium did not allow the pathologist to report any “free-margins”, even if the tumor did not invade the extra-pericardial structures.

4. Please expand a little bit more the description of imaging features of tumors. In your schematic literature review add some imaging data (e.g. tumor dimensions).

We added information also in the text; plus, the “Size” column in the table: “While the echocardiography usually highlights pericardial effusion and a solid, well-defined mass, the imaging characteristic on CT and MRI are variable. Authors reported masses with ill-defined margins, invading in the surround structures, with homogeneous or heterogeneous aspect on post contrast enhancement sequences. On MRI, a slightly hyper-T2 signal intensity was observed and reported by two papers, including us.”

5. In figure legends add more details about the images showed. Always cite the spatial plane (coronal, sagittal, axial), and exact sequences for MRI (e.g. T1w with contrast media).

Thank you very much for this comment, we apologies for this and we corrected the figure legends.

Round 2

Reviewer 1 Report

The Authors made good efforts in the attempt to ameliorate their paper.

Still believe that the paper must be checked by an english native speaker.

Author Response

Thank you very much for your time and comments. Our paper was now revised by an international C1 speaker of Medical English and changes were made.

We hope our paper rises to your expectations and meet the Journal criteria for publication.

Reviewer 2 Report

Dear authors thanks for the revisions made.

Author Response

Thank you very much for your time and appreciation.